# Mortality and Major Cardiovascular Events among Patients with Multiple Myeloma: Analysis from a Nationwide French Medical Information Database

**DOI:** 10.3390/cancers14133049

**Published:** 2022-06-21

**Authors:** Yves Cottin, Mathieu Boulin, Clara Doisy, Morgane Mounier, Denis Caillot, Marie Lorraine Chretien, Alexandre Bodin, Julien Herbert, Bernard Bonnotte, Marianne Zeller, Marc Maynadié, Laurent Fauchier

**Affiliations:** 1Cardiology Department, University Hospital, 21000 Dijon, France; yves.cottin@chu-dijon.fr; 2Pharmacy Department, EPICAD LNC UMR 1231, University of Bourgogne Franche Comté, 21000 Dijon, France; 3Registre des Hémopathies Malignes de Côte d’Or, U1231, University Hospital Center, 21000 Dijon, France; clara.doisy@chu-dijon.fr (C.D.); morgane.mounier@u-bourgogne.fr (M.M.); marc.maynadie@u-bourgogne.fr (M.M.); 4Department of Internal Medicine, University Hospital, 21000 Dijon, France; denis.caillot@chu-dijon.fr (D.C.); marie-lorraine.chretien@chu-dijon.fr (M.L.C.); bernard.bonnotte@chu-dijon.fr (B.B.); 5Cardiology Department, Trousseau Hospital and University François Rabelais, 37000 Tours, France; a.bodin@chu-tours.fr (A.B.); j.herbert@chu-tours.fr (J.H.); laurent.fauchier@chu-tours.fr (L.F.); 6PEC2, EA 7460, University of Bourgogne Franche Comté, 21000 Dijon, France; marianne.zeller@u-bourgogne.fr

**Keywords:** multiple myeloma, myocardial infarction, ischaemic stroke, bleeding, mortality

## Abstract

**Simple Summary:**

No robust data exist on the cardiovascular risks of multiple myeloma (MM) patients. We used the French nationwide hospitalization database to assess the risk of all-cause death and cardiovascular events in unselected MM patients. We demonstrated that MM patients had a higher risk of all-cause death but that they did not have a higher risk of cardiovascular death. MM patients had a lower risk of both myocardial infarction and ischaemic stroke. Conversely, they had a higher risk of major and intracranial bleedings.

**Abstract:**

Background: No robust data assesses the risk of all-cause death and cardiovascular (CV) events in multiple myeloma (MM) patients. Patients and Methods: From 1 January to 31 December 2013, 3,381,472 adults were hospitalised (for any reason) in French hospitals. We identified 15,774 patients diagnosed with known MM at baseline. The outcome analysis (all-cause death, CV death, myocardial infarction (MI), ischaemic stroke, or hospitalization for bleedings) was performed with follow-ups starting at the time of the last event. For each MM patient, a propensity score-matched patient without MM was selected. Results: The mean follow-up in the propensity-score-matched population was 3.7 ± 2.3 years. Matched patients with MM had a higher risk of all-death (yearly rate 20.02 vs. 11.39%) than patients without MM. No difference was observed between the MM group and no-MM group for CV death (yearly rate 2.00 vs. 2.02%). The incidence rate of MI and stroke was lower in the MM group: 0.86 vs. 0.97%/y and 0.85 vs. 1.10%/y, respectively. In contrast, MM patients had a higher incidence rate of rehospitalization for major bleeding (3.61 vs. 2.24%/y) and intracranial bleeding (1.03 vs. 0.84%/y). Conclusions: From a large nationwide database, we demonstrated that MM patients do not have a higher risk of CV death or even a lower risk of both MI and ischaemic stroke. Conversely, MM patients had a higher risk of both major and intracranial bleedings, highlighting the key issue of thromboprophylaxis in these patients.

## 1. Introduction

Multiple myeloma (MM) is a plasma cell malignancy that accounts for 10% of haematological cancers. It predominantly affects older adults with a median age of 70 years at diagnosis. Cardiovascular disease (CVD) in patients with MM may derive from multiple factors unrelated to MM (age, diabetes, dyslipidaemia, obesity, prior CV diseases) and/or related to MM (cardiac AL-amyloidosis, hyperviscosity, anaemia, renal dysfunction, etc.), and/or related to MM treatment (anthracyclines, corticosteroids, alkylating agents, immunomodulatory drugs, proteasome inhibitors) [1]. In a retrospective cohort, Armenian et al. showed that MM survivors had significantly higher CVD risk compared to controls without cancer [2]. In this study, CVD included ischaemic heart diseases, stroke, and heart failure. A recent meta-analysis on carfilzomib adverse events found that the incidence of all-grade and grade ≥3 CV events was 18.1% and 8.2%, respectively [3]. Other CV data in MM patients derived from randomised clinical trials (RCTs) are not generalizable to unselected patients managed in the healthcare system at a nationwide level. Due to the design of these RCTs, only short-term CV events were collected with the heterogeneous definition of CV events. Recently, dose reductions and the implementation of guidelines for the prevention and management of CV risks may have improved MM prognosis [4,5,6]. Finally, no large and recent data exist on CV risk in MM patients.

Based on a French nationwide hospitalization database, we aimed to assess the risk of all-cause death and CV outcomes in unselected MM patients of daily clinical practices.

## 2. Methods

### 2.1. Study Design

This longitudinal cohort study was based on the national hospitalization database covering hospital care for the entire French population. Data for all patients admitted to French hospitals from January to December 2013 with at least five years of follow-up (or until death) were collected from the national administrative PMSI (*Programme de Médicalisation des Systémes d’Information*) database, which was inspired by the US Medicare system. It covers more than 98% of the French population (67 million people) from birth (or immigration) to death (or emigration), even if a person changes occupations or retires. The PMSI contains individual anonymised information on each hospitalization, linked to create a longitudinal record of hospital stays and diagnoses for each patient. The reliability of PMSI data has already been assessed and this database has been used to study patients with CV conditions [7,8,9,10].

The study was conducted retrospectively without any impact on patient care. Ethical approval was not required as all data were anonymised. The French Data Protection Authority granted access to the PMSI data. Procedures for data collection and management were approved by the Commission Nationale de l’Informatique et des Libertés (CNIL), the independent National Ethical Committee protecting human rights in France, which ensures that all information is kept confidential and anonymous, in compliance with the Declaration of Helsinki (authorization number 1897139).

### 2.2. Study Population

From 1 January 2013 to 31 December 2013, a total of 3,381,472 adults (aged ≥18 years) were hospitalised in France and had at least five years of complete follow-up (or until death). For each hospital stay, combined diagnoses at discharge were obtained. Each variable was identified using ICD-10 codes, and MM was identified with the following ICD-10 codes: C88 and C90. Exclusion criteria were age <18 years. Patients without MM at baseline but with MM occurrence during follow-up were excluded from the analysis (Figure 1).

### 2.3. Outcomes

Patients were followed until 31 December 2019. The endpoints were evaluated with follow-up starting from the date of the first hospitalization in 2013 until the date of each specific outcome or the date of the last follow-up in the absence of any outcome. All-cause death, CV death, MI, ischaemic stroke, major bleeding, and intracranial bleeding were identified using the respective ICD-10 or procedure codes. Major bleeding was defined using the Bleeding Academic Research Consortium (BARC) definitions [11]. The cause of death (all-cause or CV) was identified based on the main diagnosis during the hospitalization resulting in death.

### 2.4. Statistical Analysis

Qualitative variables were described using frequencies and quantitative variables using means ± standard deviations (SD). A multivariate analysis for clinical outcomes during the whole follow-up in the group of interests was performed using a Cox proportional hazards regression model. Hazard ratios (HR) with two-sided 95% confidence intervals (CI) were also estimated using the model by Fine and Gray for competing risks for (1) CV and non-CV deaths, (2) MI and all-cause deaths, and (3) ischaemic stroke and all-cause deaths.

Owing to the non-randomised nature of the study, a propensity-score matching was used to limit the potential confounders in the treatment–outcomes relationship. For each patient with MM, a propensity score-matched patient without MM was selected (1:1) using the one-to-one nearest neighbour method and no replacement. We assessed the distribution of demographic data and comorbidities in the two cohorts with standardised differences, which were calculated as the difference in the means or proportions of a variable divided by a pooled estimate of the SD of the variable. A standardised difference of 5% or less indicated a negligible difference between the means of the two cohorts Appendix A.

All comparisons with *p* < 0.05 were considered statistically significant. All analyses were performed using Enterprise Guide 7.1 (SAS Institute, Inc., Cary, NC, USA) and STATA version 12.0 (Stata Corp, College Station, TX, USA).

## 3. Results

### 3.1. Baseline Unmatched Population

Over the study period, 3,381,472 patients were included in the study. Among them, 15,774 had a history of MM (Figure 1). MM patients were more likely to be males, older, and more likely to have hypertension and other cardiovascular diseases (Table 1).

In the matched population, most baseline characteristics were similar for the two groups except for ischaemic heart disease and valve disease, which were more frequent in MM patients (Table 2).

During the follow-up of 4.7 ± 1.8 years (median 5.4, IQR 5.0–5.8 years), all cause-deaths, CV deaths, and other CV events were more frequent in MM patients Appendix A.

### 3.2. Outcomes in the Matched Cohort

The mean follow-up in the propensity-score-matched population was 3.7 ± 2.3 years (median 5.0, IQR 1.3–5.7 years). Incident outcomes in the matched population and hazard ratio associated with MM vs. without are presented in Table 3 and Table 4, respectively.

The risk of all-cause deaths was almost two times higher in MM patients, with an incidence rate of 20.02 vs. 11.39%/y (Table 3 and Figure 2). No significant differences were observed between the two groups in terms of CV death (Table 3 and Table 4 and Figure 2).

Incidence rates of MI and ischaemic stroke were significantly lower in MM patients: 0.86 vs. 0.97%/y; *p* = 0.03 and 0.85 vs. 1.10%/y; *p* < 0.0001, respectively (Table 3 and Figure 3). In contrast, MM patients had a higher incidence rate of hospitalization for major and intracranial bleedings: 3.61 vs. 2.24%/y; *p* < 0.0001 and 1.03 vs. 0.84%/y; *p* = 0.0005, respectively (Table 3 and Figure 4). The competing risk analysis confirmed the lower risks of both MI and ischaemic stroke and the higher risks of both major and intracranial bleedings in MM patients (Table 4).

Results were similar in the sensitivity analysis limited to MM patients recently diagnosed (diagnosis within the three previous months) Appendix A.

## 4. Discussion

In this large nationwide retrospective cohort study, we showed that MM patients had a higher incidence of all-cause death, a lower incidence of both MI and ischaemic stroke, and a higher incidence of both major and intracranial bleedings than patients without MM. No differences were observed for CV-related deaths. Our results were confirmed in the analysis restricted to patients with a recent MM diagnosis and in a competing risk analysis. The competing risk analysis is a major strength of our study considering the higher risk of non-cardiovascular death in MM patients [12,13].

We confirmed the poor prognosis of MM in the general population even if proteasome inhibitors and immunomodulatory agents (used worldwide since 2000) explain the overall survival improvement observed from large US databases [12,14]. An analysis that included 90,975 MM patients from the Surveillance Epidemiology and End Results (SEER) database showed a reduction in all-cause mortality at six months from 1974 to 2014, and notably from 2006 [12]. In the future, monoclonal antibodies and chimeric antigen receptor (CAR) T cell therapy will again improve responses and survival in MM patients. Considering the immunoediting of MM cells and the great immunosuppressive impact of the bone marrow MM microenvironment, the optimal combination of anti-MM therapies as well as timing and treatment duration represent important factors to be addressed in future MM trials [15]. Regardless of the well-known or putative mechanisms of all of these anti-MM recent therapies, their immunosuppressive, immunomodulatory, and anti-angiogenic effects may potentially induce benefits outside the MM or conversely severe adverse events [15].

To date, the most common causes of death in MM patients are the disease itself, followed by CV events, infections, and kidney failure [12,13]. Overall MM mortality declined sharply over time due to declines in MM- and CV-related mortality, according to SEER data [13]. In the cohort, CV mortality decreased significantly from 12.6 per patient-year (1995–1999) to 12.3 per patient-year (2000–2004) and 9.1 per patient-year (2005–2009) (*p* < 0.0001). During the mean follow-up of 81 months of the 3897 MM patients, the first cause of death was MM followed by CV events; 91.7 and 9.1 per patient-year, respectively. We observed the same trends with yearly rates of 20.02 and 2.00% for all-cause death and CV death in our MM patients, respectively. A retrospective analysis of 3954 MM patients included in phase 2/3 RCTs testing bortezomib reported an incidence of CV death from 0.6 to 1.4%/y [16]. This lower incidence rate may be explained by the retrospective analysis using patient-level data from selected populations. Our real-life study included unselected older patients with higher baseline CV risks and comorbidities [17].

The absence of difference in terms of CV mortality between matched patients with and without MM is very encouraging. More interestingly, MM was associated with a lower risk of both MI and stroke. It seems to be very hypothetical that anti-MM therapies (mainly proteasome inhibitors and immunomodulatory agents widely used at the time of the study period) may have arterial benefits in this high-risk vascular category of older patients with frequent comorbidities. Conversely, SEER data demonstrated that carfilzomib use—a second-generation, highly selective, and irreversible proteasome inhibitor—was significantly associated with an increased risk of heart failure (HR 1.47, *p* = 0.0002), ischaemic heart disease (HR 1.45, *p* = 0.0002), and hypertension (HR 3.33, *p* < 0.0001), whereas there was no association between its use and cardiac conduction disorders (arrhythmia and heart blocks) [18]. Therefore, we thought that the benefit observed in MM patients is due to intensive management for the primary and secondary preventions of atherothrombotic diseases. MM-related venous thromboembolism (VTE) is very well described for several reasons: patients are older, with age-related risks of VTE; cancer (MM in particular) multiplies the VTE risk; above all, the VTE rate of 4.1 per 100 patient-months was associated with immunomodulatory drug intake without any pharmacologic VTE prophylaxis [19]. This was highlighted with thalidomide, lenalidomide, and pomalidomide, the three available drugs that were widely prescribed in approximately all MM patients during the study period. It is even mandatory to associate pharmacological thromboprophylaxis in patients receiving immunomodulatory agents [20]. Furthermore, MM patients have iteratively scheduled visits due to the severity of the disease and/or to receive anti-MM treatment. At each visit, a clinical and biological assessment, including a medication review, is usually performed by the haematologist. This “intensive” follow-up may be beneficial for the patient’s CV health, considering that a cardiologist is routinely consulted before initiating anti-MM cardiotoxic agents. Finally, the probability that a patient receives optimal antithrombotic therapy over prolonged periods is high.

Conversely, the increase in both major and intracranial bleedings is probably due to this intensive anti-MM thromboprophylaxis in older adults. Even with therapeutic re-evaluations during visits, patients may have concomitantly received one anticoagulant and one antiplatelet agent over too long of a period as they may have received an anticoagulant after immunomodulatory discontinuation. Coagulation disorders are frequent and remain major clinical challenges in MM patients [21,22]. The underlying mechanisms for bleeding events are multiple and are poorly correlated with initial anti-MM regimens [23]. Hinterleitner et al. demonstrated that bleedings in MM were predominantly caused by deficiencies in primary haemostasis associated with disease progression [24].

## 5. Limitations

The main limitation is inherent to the retrospective, observational nature of the study and its potential biases. Diagnoses and the occurrence of outcomes were based on diagnostic codes registered by a physician and were not further checked externally [10]. We had no information of death occurring outside hospitals in the database. Our large population of patients hospitalised with MM likely represented a heterogeneous group of patients admitted with various kinds of illnesses and severities, which may have affected prognosis.

Another limitation is the lack of information on MM treatments or other cardiovascular agents and their possible changes over time. Furthermore, the non-randomised design of the analysis leaves a risk of residual confounding factors. Definitive conclusions between groups may not be fully appropriate even though multivariable matching was performed. However, it cannot fully eradicate the possible confounding variables between groups with and without MM.

## 6. Conclusions

In our large cohort of unselected French patients, we found that MM patients had a higher risk of all-cause mortality and major and intracranial bleedings than patients without MM. Conversely, MM patients had a lower risk of both MI and ischaemic stroke. In MM patients, thromboprophylaxis should be carefully evaluated in order to properly balance the risks of bleeding and ischemia. Better knowledge of newer anti-MM therapy effects on MM cells, microenvironments, endothelial cells, and vessels offers encouraging perspectives for individualised approaches to increase responses and, at the same time, decrease CV events and thromboprophylaxis complications.

## Figures and Tables

**Figure 1 cancers-14-03049-f001:**
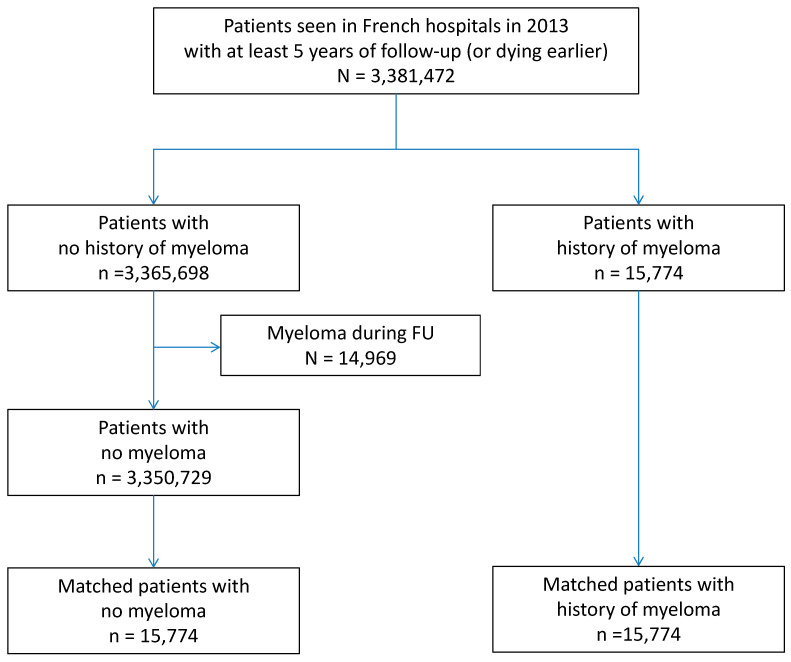
Flow chart of the study describing the incidences of major CV events in patients with or without a history of myeloma seen in French hospitals in 2013, with at least five years of follow-up. (FU = follow-up).

**Figure 2 cancers-14-03049-f002:**
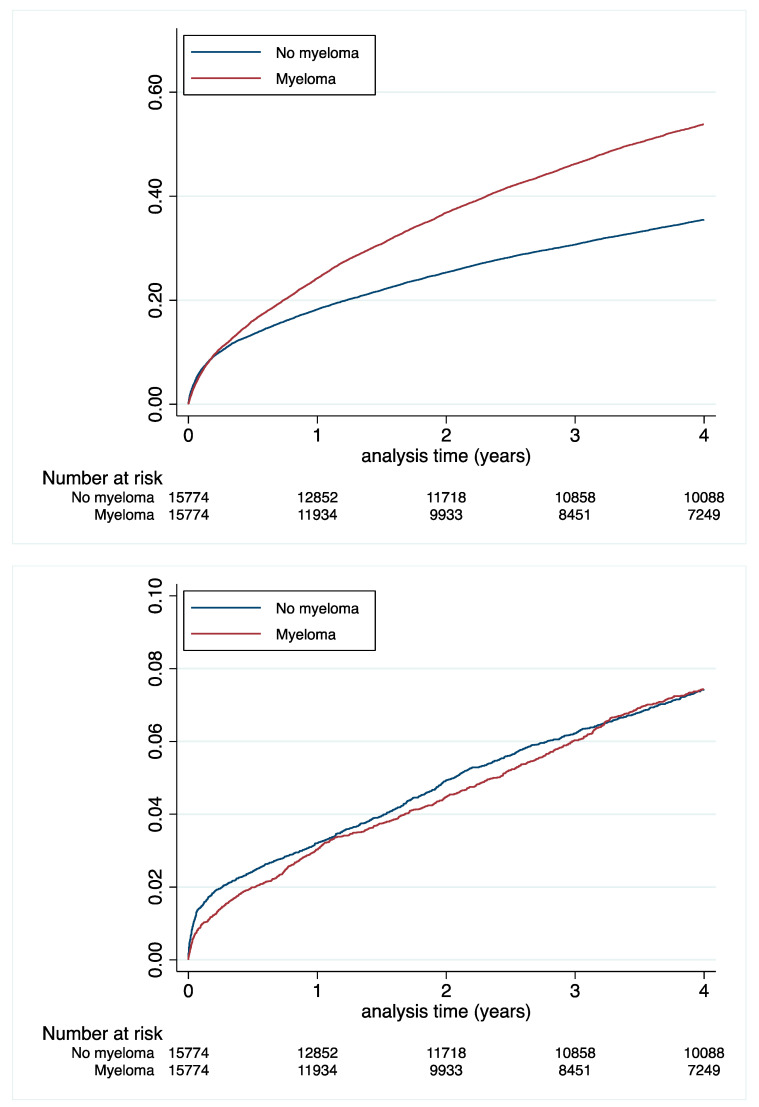
Cumulative incidences of all-cause death (**top** panel) or cardiovascular death (**lower** panel) in the matched population.

**Figure 3 cancers-14-03049-f003:**
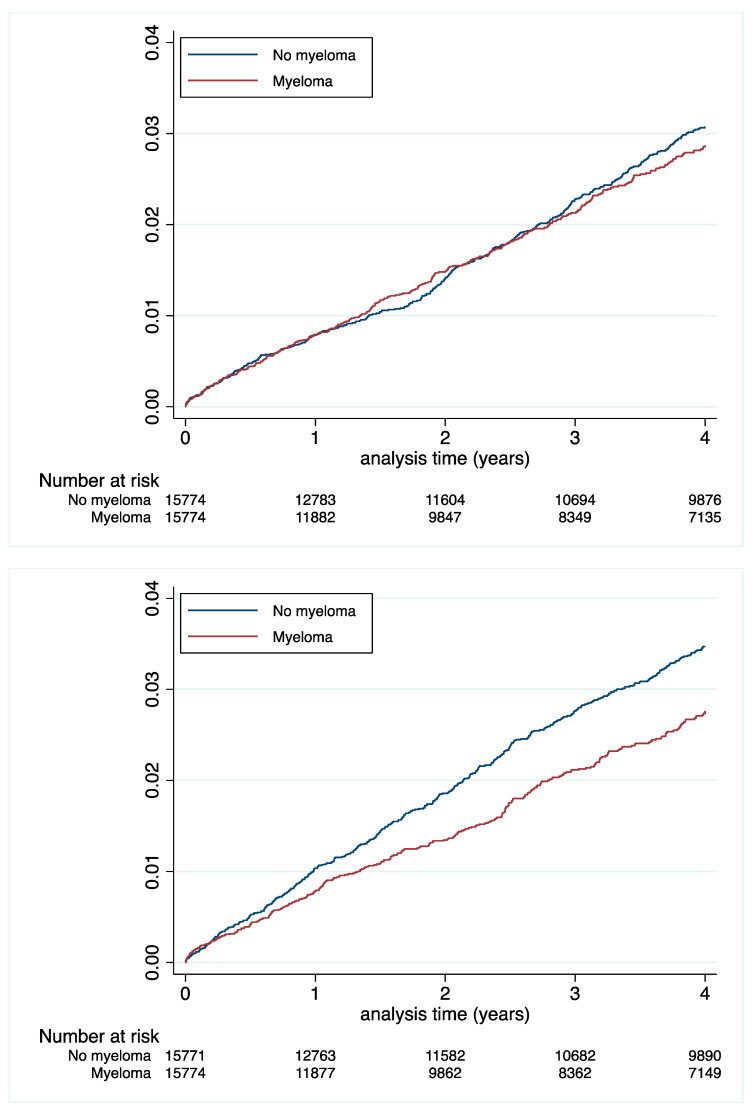
Cumulative incidences of myocardial infarction (**top** panel) and ischaemic stroke (**lower** panel) in the matched population.

**Figure 4 cancers-14-03049-f004:**
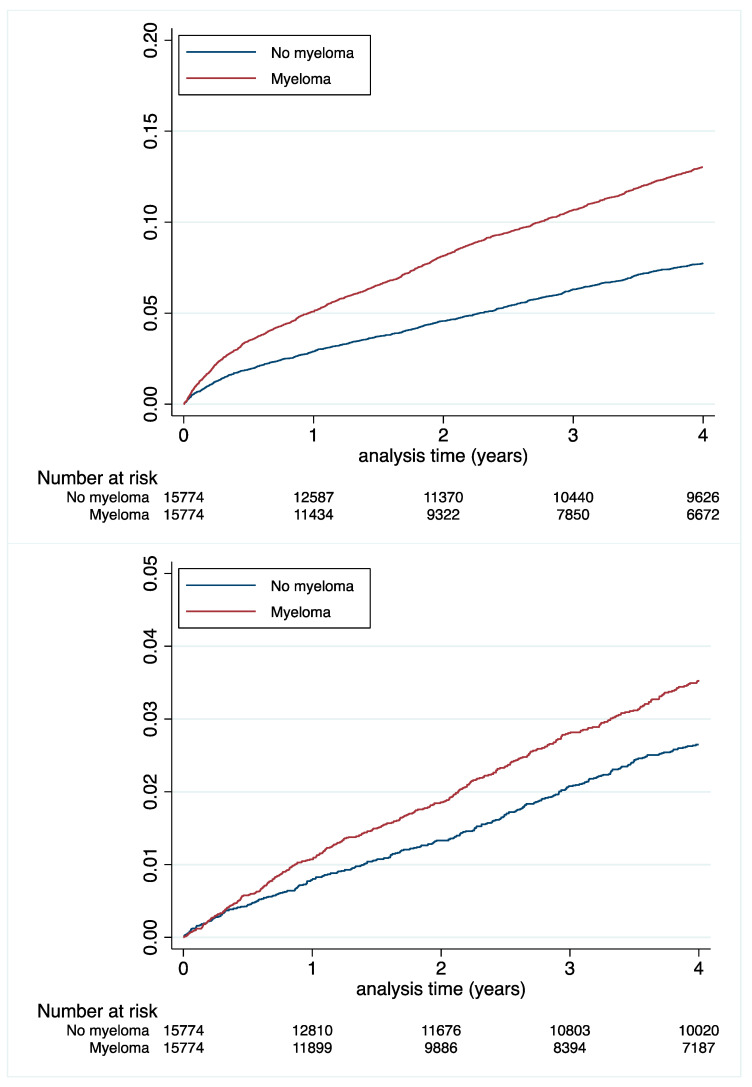
Cumulative incidences of major bleeding (**top** panel) and intracranial bleeding (**lower** panel) in the matched population.

**Table 1 cancers-14-03049-t001:** Baseline characteristics of the unmatched population *.

	Without Multiple Myeloma	With Multiple Myeloma	*p*	Total
	(*n* = 3,350,729)	(*n* = 15,774)		(*n* = 3,366,503)
Age, years	59.1 ± 21.5	71.2 ± 11.6	<0.0001	59.2 ± 21.5
Sex (male)	1,568,467 (46.8)	8686 (55.1)	<0.0001	1,577,153 (46.8)
Hypertension	1,022,172 (30.5)	6293 (39.9)	<0.0001	1,028,465 (30.5)
Diabetes mellitus	465,033 (13.9)	2209 (14.0)	0.65	467,242 (13.9)
Heart failure	351,359 (10.5)	2655 (16.8)	<0.0001	354,014 (10.5)
History of pulmonary oedema	25,916 (0.8)	237 (1.5)	<0.0001	26,153 (0.8)
Valve disease	120,980 (3.6)	801 (5.1)	<0.0001	121,781 (3.6)
Previous endocarditis	4486 (0.1)	56 (0.4)	<0.0001	4542 (0.1)
Dilated cardiomyopathy	77,368 (2.3)	592 (3.8)	<0.0001	77,960 (2.3)
Coronary artery disease	357,923 (10.7)	1805 (11.4)	0.002	359,728 (10.7)
Previous myocardial infarction	57,239 (1.7)	255 (1.6)	0.38	57,494 (1.7)
Previous PCI	88,528 (2.6)	307 (1.9)	<0.0001	88,835 (2.6)
Previous CABG	12,205 (0.4)	53 (0.3)	0.56	12,258 (0.4)
Vascular disease	289,114 (8.6)	1398 (8.9)	0.3	290,512 (8.6)
Atrial fibrillation	321,479 (9.6)	2117 (13.4)	<0.0001	323,596 (9.6)
Previous pacemaker or ICD	104,089 (3.1)	500 (3.2)	0.65	104,589 (3.1)
Ischaemic stroke	63,509 (1.9)	252 (1.6)	0.01	63,761 (1.9)
Intracranial bleeding	35,056 (1.0)	155 (1.0)	0.43	35,211 (1.0)
Smoking	231,029 (6.9)	733 (4.6)	<0.0001	231,762 (6.9)
Dyslipidaemia	441,094 (13.2)	2051 (13.0)	0.55	443,145 (13.2)
Obesity	355,793 (10.6)	1237 (7.8)	<0.0001	357,030 (10.6)
Alcohol-related diagnoses	187,581 (5.6)	447 (2.8)	<0.0001	188,028 (5.6)
Chronic kidney disease	117,537 (3.5)	1707 (10.8)	<0.0001	119,244 (3.5)
Lung disease	340,348 (10.2)	1837 (11.6)	<0.0001	342,185 (10.2)
Sleep apnoea syndrome	134,202 (4.0)	508 (3.2)	<0.0001	134,710 (4.0)
COPD	185,911 (5.5)	993 (6.3)	<0.0001	186,904 (5.6)
Liver disease	114,867 (3.4)	586 (3.7)	0.05	115,453 (3.4)
Thyroid diseases	182,181 (5.4)	1112 (7.0)	<0.0001	183,293 (5.4)
Inflammatory disease	176,442 (5.3)	1291 (8.2)	<0.0001	177,733 (5.3)
Anaemia	272,393 (8.1)	6489 (41.1)	<0.0001	278,882 (8.3)
Previous cancer	486,308 (14.5)	15,774 (100.0)	<0.0001	502,082 (14.9)
Poor nutrition	127,872 (3.8)	1752 (11.1)	<0.0001	129,624 (3.9)
Cognitive impairment	114,381 (3.4)	604 (3.8)	0.004	114,985 (3.4)
Illicit drug use	13,618 (0.4)	23 (0.1)	<0.0001	13,641 (0.4)

* The mean follow-up was 4.7 ± 1.8 years (median 5.4, IQR 5.0–5.8 years). Values are *n* (%) or mean ± SD. CABG = coronary artery bypass graft; COPD = chronic obstructive pulmonary disease; PCI = percutaneous coronary intervention; SD = standard deviation.

**Table 2 cancers-14-03049-t002:** Baseline characteristics in the matched population.

	Without Multiple Myeloma	With Multiple Myeloma	*p*	Total
	(*n* = 15,774)	(*n* = 15,774)		(*n* = 31,548)
Age, years	71.4 ± 11.8	71.2 ± 11.6	0.17	71.3 ± 11.7
Gender (male)	8603 (54.5)	8686 (55.1)	0.35	17,289 (54.8)
Hypertension	6329 (40.1)	6293 (39.9)	0.68	12,622 (40.0)
Diabetes mellitus	2194 (13.9)	2209 (14.0)	0.81	4403 (14.0)
Heart failure	2621 (16.6)	2655 (16.8)	0.61	5276 (16.7)
History of pulmonary oedema	230 (1.5)	237 (1.5)	0.74	467 (1.5)
Valve disease	1005 (6.4)	801 (5.1)	<0.0001	1806 (5.7)
Previous endocarditis	57 (0.4)	56 (0.4)	0.92	113 (0.4)
Dilated cardiomyopathy	533 (3.4)	592 (3.8)	0.07	1125 (3.6)
Coronary artery disease	2324 (14.7)	1805 (11.4)	<0.0001	4129 (13.1)
Previous myocardial infarction	383 (2.4)	255 (1.6)	<0.0001	638 (2.0)
Previous PCI	492 (3.1)	307 (1.9)	<0.0001	799 (2.5)
Previous CABG	108 (0.7)	53 (0.3)	<0.0001	161 (0.5)
Vascular disease	1920 (12.2)	1398 (8.9)	<0.0001	3318 (10.5)
Atrial fibrillation	2494 (15.8)	2117 (13.4)	<0.0001	4611 (14.6)
Previous pacemaker or ICD	836 (5.3)	500 (3.2)	<0.0001	1336 (4.2)
Ischaemic stroke	243 (1.5)	252 (1.6)	0.68	495 (1.6)
Intracranial bleeding	131 (0.8)	155 (1.0)	0.15	286 (0.9)
Smoker	723 (4.6)	733 (4.6)	0.79	1456 (4.6)
Dyslipidaemia	2039 (12.9)	2051 (13.0)	0.84	4090 (13.0)
Obesity	1225 (7.8)	1237 (7.8)	0.8	2462 (7.8)
Alcohol-related diagnoses	444 (2.8)	447 (2.8)	0.92	891 (2.8)
Chronic kidney disease	1652 (10.5)	1707 (10.8)	0.32	3359 (10.6)
Lung disease	1775 (11.3)	1837 (11.6)	0.27	3612 (11.4)
Sleep apnoea syndrome	477 (3.0)	508 (3.2)	0.32	985 (3.1)
COPD	945 (6.0)	993 (6.3)	0.26	1938 (6.1)
Liver disease	546 (3.5)	586 (3.7)	0.23	1132 (3.6)
Thyroid diseases	1116 (7.1)	1112 (7.0)	0.93	2228 (7.1)
Inflammatory disease	1209 (7.7)	1291 (8.2)	0.09	2500 (7.9)
Anaemia	6477 (41.1)	6489 (41.1)	0.89	12,966 (41.1)
Previous cancer	4261 (27.0)	15,774 (100.0)	<0.0001	20,035 (63.5)
Poor nutrition	1656 (10.5)	1752 (11.1)	0.08	3408 (10.8)
Cognitive impairment	610 (3.9)	604 (3.8)	0.86	1214 (3.8)
Illicit drug use	17 (0.1)	23 (0.1)	0.34	40 (0.1)

Values are *n* (%) or mean ±SD. CABG = coronary artery bypass graft; COPD = chronic obstructive pulmonary disease; PCI = percutaneous coronary intervention; SD = standard deviation.

**Table 3 cancers-14-03049-t003:** Incident outcomes in the matched population *.

	Without Multiple Myeloma (*n* = 15,774)		With Multiple Myeloma (*n* = 15,774)		*p*
	Number of Events	Incidence, %/y (95% CI)	Number of Events	Incidence, %/y (95% CI)	
All-cause death	7232	11.39 (11.13–11.65)	10,524	20.02 (19.65–20.41)	<0.0001
Cardiovascular death	1285	2.02 (1.92–2.14)	1053	2.00 (1.89–2.13)	0.41
Myocardial infarction	608	0.97 (0.90–1.05)	449	0.86 (0.79–0.95)	0.03
Ischaemic stroke	686	1.10 (1.02–1.18)	440	0.85 (0.77–0.93)	<0.0001
Major bleeding	1371	2.24 (2.12–2.36)	1784	3.61 (3.44–3.78)	<0.0001
Intracranial bleeding	531	0.84 (0.77–0.92)	539	1.03 (0.95–1.12)	0.0005

* The mean follow-up was 4.7 ± 1.8 years (median 5.4, IQR 5.0-5.8 years).

**Table 4 cancers-14-03049-t004:** Hazard ratios associated with multiple myeloma (vs without) for incident outcomes.

	Model A	Model B	Model C	Model D
All-cause death	2.837 (2.783–2.892)	2.092 (2.052–2.133)	1.781 (1.747–1.816)	1.658 (1.609–1.708)
Cardiovascular death	1.380 (1.299–1.466)	1.010 (0.951–1.073)	0.958 (0.902–1.018)	0.928 (0.855–1.007) *
Myocardial infarction	1.206 (1.099–1.323)	0.909 (0.829–0.998)	0.924 (0.842–1.014)	0.871 (0.771–0.985) †
Ischaemic stroke	1.094 (0.997–1.202)	0.821 (0.747–0.901)	0.814 (0.741–0.894)	0.745 (0.660–0.840) ‡
Major bleeding	2.088 (1.993–2.188)	1.581 (1.509–1.656)	1.685 (1.608–1.766)	1.535 (1.431–1.647)
Intracranial bleeding	1.852 (1.701–2.015)	1.414 (1.299–1.539)	1.336 (1.227–1.454)	1.195 (1.060–1.348)

Hazard ratios are presented with 95% confidence intervals in parentheses. Model A—unadjusted. Model B—adjusted for age and sex. Model C—adjusted for all risk factors and non-cardiovascular comorbidities: age, sex, hypertension, smoker, dyslipidaemia, diabetes mellitus, obesity, alcohol-related diagnoses, previous ischaemic stroke, intracranial bleeding, chronic kidney disease, lung disease, sleep apnoea syndrome, chronic obstructive pulmonary disease, liver disease, gastroesophageal reflux, thyroid diseases, inflammatory disease, anaemia, poor nutrition, cognitive impairment, and illicit drug use. Model D--propensity score-matched analysis adjusted for variables mentioned for Model C. * Hazard ratio = 0.794 (0.732-0.862), *p* < 0.0001, by the Fine and Gray model for competing risks of cardiovascular and non-cardiovascular death. † Hazard ratio = 0.711 (0.629–0.804), *p* < 0.0001, by the Fine and Gray model for competing risks of myocardial infarction and all-cause death. ‡ Hazard ratio = 0.612 (0.543–0.690), *p* = 0.009, by the Fine and Gray model for competing risks of ischaemic stroke and all-cause death.

## Data Availability

The data presented in this study are available on request from the corresponding author. The data are not publicly available due to University Hospital of Dijon and Tours property rules.

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
