# Peer review of "Mortality and Major Cardiovascular Events among Patients with Multiple Myeloma: Analysis from a Nationwide French Medical Information Database"

_cancers, 2022, doi:10.3390/cancers14133049_

Round 1

Reviewer 1 Report

Summary- a bit confusing to read. Suggest modify

Abstract: Suggest another sentence to say why you did this study

Conclusions line 36- had a higher risk of major and …

Methods Overall- Please try and write more concisely. Level of detail is not always necessary especially when references are provided.

There is no need to list all covariates in Cox model ( could just refer to Table 1).

Similarly in Results- Please write more concisely. There is no need to present all results in text as are there in Tables. You could say for example MM patients were more likely to be males, older age groups and more likely to have hypertension and cardiovascular disease (Table 1).

Discussion needs to be far more focused. There is no need to give so much information especially on other papers – it just makes it harder to read and understand the main messages of this paper.

I would suggest a paragraph focussing on the key messages of the paper, possible relevance to MM patients and any suggestions for implications of this paper.

Author Response

Summary- a bit confusing to read. Suggest modify

Response: You are right. We have modified the Summary as follows: « No robust data exist on the cardiovascular risk of multiple myeloma (MM) patients. We used the French nationwide hospitalization database to assess the risk of all-cause death and of cardiovascular events in unselected MM patients. We demonstrated that MM patients have a higher risk of all-cause death but that they do not have any higher risk of cardiovascular death. MM patients had a lower risk of both myocardial infarction and ischemic stroke. Conversely, they had a higher risk of both major and intracranial bleedings ».

Abstract: Suggest another sentence to say why you did this study

Conclusions line 36- had a higher risk of major and …

Response: We changed the abstract as follows: « Background: No robust data assessed the risk of all-cause death and of cardiovascular (CV) events in multiple myeloma (MM) patients. » and « Conclusions: From a large nationwide database, we demonstrated that MM patients do not have any higher risk of CV death and even a lower risk of both MI and ischemic stroke. Conversely, MM patients had a higher risk of both major and intracranial bleedings, highlighting the key issue of thromboprophylaxis in these patients ».

Methods Overall- Please try and write more concisely. Level of detail is not always necessary especially when references are provided.

There is no need to list all covariates in Cox model ( could just refer to Table 1).

Response: As recommended, we have widely shortened the Methods section. As a first example, the sentence : «Major bleeding was defined using the Bleeding Academic Research Consortium (BARC) definitions as bleeding with a reduction in the hemoglobin level of at least 20g per liter, or with transfusion of at least 1 unit of blood, or symptomatic bleeding in a critical area or organ (e.g., intracranial, intraspinal, intraocular, retroperitoneal, intra-articular or pericardial, or intramuscular with compartment syndrome) or bleeding that causes death.11 » was modified as follows:  « Major bleeding was defined using the Bleeding Academic Research Consortium (BARC) definitions.11 » 

As a second example, the sentence : «A multivariate analysis for clinical outcomes during the whole follow-up in the groups of interests was performed using a Cox proportional hazards regression model including baseline characteristics (age, sex, hypertension, smoking, dyslipidemia, diabetes mellitus, obesity, alcohol-related diagnoses, previous ischemic stroke, intracranial bleeding, chronic kidney disease, lung disease, sleep apnoea syndrome, chronic obstructive pulmonary disease, liver disease, gastroesophageal reflux, thyroid diseases, inflammatory disease, anaemia, poor nutrition, cognitive impairment, and illicit drug use) and reporting hazard ratio (HR) with two-sided 95% confidence intervals (CI). » was modified as follows:  «A multivariate analysis for clinical outcomes during the whole follow-up in the groups of interests was performed using a Cox proportional hazards regression model.» 

Similarly in Results- Please write more concisely. There is no need to present all results in text as are there in Tables. You could say for example MM patients were more likely to be males, older age groups and more likely to have hypertension and cardiovascular disease (Table 1).

Response: You are right. As for Methods, we have now shortened the Results section. The majority of or results are presented in Tables and Figures or in Supplementary Tables.

Discussion needs to be far more focused. There is no need to give so much information especially on other papers – it just makes it harder to read and understand the main messages of this paper.

I would suggest a paragraph focussing on the key messages of the paper, possible relevance to MM patients and any suggestions for implications of this paper.

Response: Reviewer 2 made a close comment concerning discussion elements. Thus, we have deeply modified all the Discussion section. As recommended, we have deleted several sentences comparing data from other studies and associated references. Main messages of the work (lower risk of MI and stroke, higher bleedings risk) have been reinforced and discussed regarding overall MM management including novel therapeutics and their potential impact for the daily clinical practice. 

Reviewer 2 Report

Yves Cottin et al. uncovered 

Mortality and major cardiovascular events among patients with 

multiple myeloma.

Points to be considered:

1. I would suggest to restructure the manuscript as follows:

P (Patient, population or problem)

Who or what is the patient, population or problem in question?

I (Intervention)

What is the intervention (action or treatment) being considered?

C (Comparison or control)

What other interventions should be considered?

O (Outcome or objective)

What is the desired or expected outcome or objective?

T (Time frame/treatment)

How long will it take to reach the desired outcome? AND/OR STUDy (picos)

2. The authors highlight the role of anticoagulation for MM management. Indeed, Venous thromboembolism (VTE) is a disease with a high prevalence in elderly people, affecting > 5% of the population > 65 years of age. Cancer patients have a 4.3-fold higher incidence of thrombotic complications, due to multiple risk factors that are not always related to the disease. Among hematologic malignancies, multiple myeloma (MM) confers a high risk of developing such complications, with a VTE rate of nearly 10%. Within this scenario, As is now well known, tumors grow and evolve through a constant crosstalk with the surrounding microenvironment, and emerging evidence indicates that angiogenesis and immunosuppression frequently occur simultaneously in response to this crosstalk.  Endothelial cells and vessels make no exception:  The underlying message here is that more precision and individualized approaches need to be tested in well designed clinical trials – a challenge, but I would be interested in their perspective of how this might be done.

3. This reviewer personally misses some insights regarding strategies combining anti-angiogenic therapy and immunotherapy that seem to have the potential to tip the balance of the tumor microenvironment and improve treatment response and that could have improved MM outcome (please refer to PMID: 33194767 and expand.

Author Response

Yves Cottin et al. uncovered 

Mortality and major cardiovascular events among patients with 

multiple myeloma.

Points to be considered:

  1. I would suggest to restructure the manuscript as follows:

P (Patient, population or problem)

Who or what is the patient, population or problem in question?

I (Intervention)

What is the intervention (action or treatment) being considered?

C (Comparison or control)

What other interventions should be considered?

O (Outcome or objective)

What is the desired or expected outcome or objective?

T (Time frame/treatment)

How long will it take to reach the desired outcome? AND/OR STUDy (picos)

Response: Thanks for the comment. We agree that the structure of the manuscript needed several improvements, and we made several changes based on the comments by reviewer 1 and 2 in the methods and in the discussion. PICOT format is a helpful approach for summarizing research questions exploring the effect of therapy, particularly in a randomized trial. However and very respectfully, our analysis is purely observational and is not focused on therapy. It is an epidemiologic analysis of a specific disease (multiple myeloma) with a description of the current history of competing outcomes, particularly cardiovascular or non-cardiovascular death and also regarding cardiovascular events competing with all-cause death. The different comments on the possible role of therapy for explaining the results remains indirect speculations and are not the core of our analysis. We thus suggest not using the PICOT format for our manuscript since it may not be fully appropriate for our analysis, particularly regarding the I (Intervention, What is the intervention - action or treatment - being considered?) And C (Comparison or control, What other interventions should be considered?).  We hope that reviewer 2 will understand our point of view.

  1. The authors highlight the role of anticoagulation for MM management. Indeed, Venous thromboembolism (VTE) is a disease with a high prevalence in elderly people, affecting > 5% of the population > 65 years of age. Cancer patients have a 4.3-fold higher incidence of thrombotic complications, due to multiple risk factors that are not always related to the disease. Among hematologic malignancies, multiple myeloma (MM) confers a high risk of developing such complications, with a VTE rate of nearly 10%. Within this scenario, As is now well known, tumors grow and evolve through a constant crosstalk with the surrounding microenvironment, and emerging evidence indicates that angiogenesis and immunosuppression frequently occur simultaneously in response to this crosstalk.  Endothelial cells and vessels make no exception:  The underlying message here is that more precision and individualized approaches need to be tested in well designed clinical trials – a challenge, but I would be interested in their perspective of how this might be done.

Response: We agree with your comment. In accordance with Reviewer 1 comments and yours, we have widely modified our discussion. In particular, we have detailed hypotheses to explain decrease in MI and ischemic stroke occurrence and increase in major and intracranial bleedings. Thromboprophylaxis is probably the key factor with potentially cardiovascular (including endothelial) effects of anti-MM therapies (proteasome inhibitors, immunomoodulatory drugs, monoclonal antibodies…).

  1. This reviewer personally misses some insights regarding strategies combining anti-angiogenic therapy and immunotherapy that seem to have the potential to tip the balance of the tumor microenvironment and improve treatment response and that could have improved MM outcome (please refer to PMID: 33194767 and expand.

 Response: You are right. Thank you for your very interesting reference we have now incorporated with others on the subject in the new version of our manuscript. As said above, the Discussion section has been widely modified.

Round 2

Reviewer 2 Report

The authors have clarified several of the questions I raised in my previous review. Most of the major problems have been addressed by this revision.